# High Iodine Urinary Concentration Is Associated with High TSH Levels but Not with Nutrition Status in Schoolchildren of Northeastern Mexico

**DOI:** 10.3390/nu13113975

**Published:** 2021-11-08

**Authors:** Aidy Gonzalez-Nunez, Pablo Garcia-Solis, Silvia G. Ramirez-Garcia, German Flores-Ramirez, Marcela Vela-Amieva, Victor J. Lara-Diaz, Augusto Rojas-Martinez

**Affiliations:** 1Tecnologico de Monterrey, Escuela de Medicina y Ciencias de la Salud, Monterrey CP 64710, Mexico; lara-diaz.vj@tec.mx; 2Laboratorio de Endocrinologia y Nutricion, Departamento de Investigacion Biomedica, Facultad de Medicina, Universidad Autonoma de Queretaro, Queretaro CP 76176, Mexico; pablo.garcia@uaq.mx; 3Gobierno del Estado de Nuevo Leon, Escuela Normal Miguel F Martinez, Monterrey CP 64000, Mexico; silvia.rmz.g@hotmail.com; 4Tecnica Medica de la Universidad Autonoma de Nuevo Leon, Monterrey CP 64640, Mexico; contacto@tuendocrinopediatra.com; 5Laboratorio de Errores Innatos del Metabolismo y Tamiz, Instituto Nacional de Pediatria, Secretaria de Salud, Mexico City CP 04530, Mexico; dravelaamieva@yahoo.com

**Keywords:** iodine, thyroid, nutrition

## Abstract

According to the Iodine Global Network, Mexico is considered a country with adequate national iodine intake (297 mg/L), but some regions have not been studied. We aimed to evaluate urinary iodine concentration (UIC) and its association with thyroid stimulating hormone (TSH) levels and the nutritional status in 307 children (aged 5 to 11 years) from three elementary schools of Monterrey, northern Mexico. UIC in spot urine samples and capillary TSH levels were measured to assess thyroid function, in addition to weight, height, body mass index (BMI), and waist circumference (WC). We found a median UIC of 442 mg/L and a significant association between UIC and TSH levels by logistic regression when data were adjusted for (1) age and sex; (2) age, sex, and WC; and (3) age, sex, and weight status. UIC values were higher in 7-year-old children compared to 11-year-old children. High prevalences of overweight/obesity (41%) and WC >90 pctl (22%) were observed. This study identified higher UIC levels in children than those previously reported in the country. The UIC showed a positive and significant correlation between TSH levels in the three models evaluated. More studies are needed to assess the causes and possible outcomes of high UIC levels.

## 1. Introduction

Iodine, a component of thyroid hormones (TH), is an essential trace element in human nutrition [1]. TH are essential for body and brain growth and development from the fetal stages up to late childhood. Any impairment in TH availability, during critical periods of central nervous system development, may induce irreversible brain damage, with mental retardation and neurological dysfunction [2]. Iodine deficiency may cause other sequelae, such as goiter, reproductive health impairment and IQ lowering [3]. Although iodine deficiency is considered to have been eradicated in the Americas and this region reports the highest proportion of households consuming iodized salt during the past two decades, 12 out of 24 countries in this area report iodine intake above the requirements [3,4]. Excessive iodine intake can also cause thyroid diseases such as hypo or hyperthyroidism [5].

According to the World Health Organization (WHO), the United Nations Children’s Emergency Fund (UNICEF), and the International Council for Control of Iodine Deficiency Disorders (ICCIDD) [6], the best population indicators of iodine nutrition status in schoolchildren are prevalence of goiter (palpation/ultrasound), median of urinary iodine concentration (UIC), and thyroglobulin (Tg), a thyroid protein, a precursor in the synthesis of thyroid hormones, that is increased in thyroid hyperplasia and goiter (characteristic of iodine deficiency); meanwhile, thyroid stimulating hormone (TSH) levels in dry blood samples, is the preferred indicator in neonates [6]. The UIC median is a useful indicator to assesses recent iodine intake at population level [6].

Referring to iodine nutrition, some food provides a good source of iodine, such as marine algae, fortified salt, fish, seafood, dairy, water, eggs, broccoli, peas or spinach [7]; but there is also food that may not favor normal iodine synthesis, for example, low-salicylate diets or nutrition based on lean meats, fruits and vegetables [7]. On the other hand, there are some substances and environmental chemicals that interact with iodine or thyroid normal function, such as perfluorooctanoic acid, perchlorate, bisphenols, triclosan and idophors [7,8]. There are few national data regarding iodine nutrition in Mexico; however, a recent study in central Mexico reported a median of UIC in schoolchildren of 428 μg/L [9], suggesting an excessive iodine intake. Mexico has a wide variety of dietary patterns that differ in each of its regions; the last Mexican National Health and Nutrition Survey (ENSANUT 2018–2019) [10] reported that schoolchildren population between 5–11 years old living in northern cities have the highest intake of iodine rich food, such as milk and eggs, being of 64.8% and 54.6% respectively. Besides this, high consumption of salty snacks by the same population was registered at 65% [10].

As is well known, iodine excess during pregnancy can induce hypothyroidism during fetal development due to a failed scape of the acute Wolff–Chaikoff effect when the hypothalamic pituitary thyroid axis is still immature [5]; this is particularly interesting in Mexico, because of its high prevalence of congenital hypothyroidism (CH). Previous studies have shown that the state of Nuevo Leon had a higher prevalence of CH compared to other states [11,12,13]. The most recent national report of CH in newborns, performed by the Secretary of Health, showed that the prevalence ratio of CH in Nuevo Leon was 10.3/10,000 births, whereas the national prevalence of CH at birth was 7.3/10,000 [13]. 

The aim of this study was to assess iodine nutrition and its association with TSH concentration in dry blood samples and nutrition status in schoolchildren from northeastern Mexico. 

## 2. Materials and Methods

### 2.1. Setting and Participants

This cross-sectional study was performed in the Metropolitan area of Monterrey, Nuevo León (northeastern Mexico). The study protocol, the assent (for children), and consent (for parents or tutors) forms were approved by the Ethics and Research committees of the School of Medicine at Tecnologico de Monterrey (code CEPYMTY, 18 February 2019). Three hundred and seven apparently healthy schoolchildren aged between 5 to 12 years were recruited from three different public elementary schools. All schoolchildren assented to participate in the study and written consent was obtained from their parents or main caregivers. The data for this research was collected between September–November 2019. Children with preexisting diseases, taking supplements with iodine or prescriptions that could interact with thyroid hormone synthesis or function were excluded.

### 2.2. Anthropometry and Biochemical Analysis

Weight, height, and waist circumference (WC) were measured at school, using standard anthropometric technique (NHANES 2017) [14]. Schoolchildren were measured without shoes; they wore light indoor clothing, and their pockets were emptied. Weight was recorded to the nearest 100 g and height to the nearest centimeter. Body mass index (BMI) was estimated for each participant and weight status was defined according to Center for Disease Control, USA (CDC) criteria using BMI-for-age CDC Growth Charts [15]. BMI <5th percentile (pctl) for underweight, ≥5th pctl to less than 85th pctl for normal weight, ≥85th pctl to less than 95th percentile for overweight, and ≥95th pctl for obese. Regarding WC, schoolchildren were classified using percentile values for Mexican American children and adolescents from the International Diabetes Federation (IDF) consensus of Metabolic Syndrome in Children and Adolescents as follows: <10, 10th–90th and ≥90th [16]. 

Spot urine samples were obtained from schoolchildren in 40 mL sterile plastic containers. Urine samples were kept at 4 °C until arriving to the laboratory, where they were stored at −80 °C. Urinary iodine concentration determinations were performed by microplate method based on the Sandell–Kholtoff reaction after sample digestion with ammonium persulfate according to Mendez et al.; 2016 [9]. Urine samples were analyzed at the Endocrinology and Nutrition Laboratory of Biomedical Research Department at School of Medicine of Autonomous University of Queretaro. The iodine nutrition status in children was determined according to WHO/UNICEF/ICCIDD criteria as follows: median UIC <100 μg/L insufficient iodine intake, median UIC 100–199 μg/L adequate iodine intake, median UIC 200–299 μg/L iodine intake above requirements, median UIC ≥ 300 μg/L excessive iodine intake [6].

Whole blood from a finger prick was spotted on Guthrie cards (Whatman 903™ GE Healthcare Ltd., Cardiff, UK). After spotting, Guthrie cards were dried at room temperature for 24 h and then stored in sealed low-density bags. The Guthrie cards were perforated in 3.2 mm diameter circles using automated instruments (Dried Blood Spot Puncher, PerkinElmer-Wallac Oy^®^, Turku, Finland). TSH concentration from dry blood samples were measured by fluorometric enzyme immunoassay (Neonatal hTSH FEIA Plus, Labsystems Diagnostics, Vantaa, Finland) at the Inborn Errors of Metabolism Laboratory, National Institute of Pediatrics (Health Secretary of Mexico) in Mexico City. Briefly, 3 mm disks from blood calibrators and controls were deposited in duplicate into microplates of 96 wells, and single 3 mm disks from each subject sample were placed into the microplate wells. Anti-TSH-HRP conjugate solution was added (200 μL), ensuring the proper soaking of each 3 mm blood disk. The plate was covered and incubated for 3 h at room temperature in dark with shaking speed of 650 rpm. The disks and liquid were removed and washed 4 × 300 μL of wash buffer, then 200 μL of substrate solution was added. The plate was covered and incubated for 60 min at room temperature in the dark shaking at 650 rpm. We added 100 μL of stop solution and finally, fluorescence was measured at excitation/emission 320/405 nm in a Fluoroskan Ascent (Labsystems, Vantaa, Finland) no later than 60 min and the results were given as quantitative values (mIU/L blood).

### 2.3. Statistical Analysis

D’Agostino and Pearson test was performed on continuous data to determine Gaussian distribution. Data is shown as median and interquartile range (p25–p75) for data with non-Gaussian distribution. Non-parametric tests were used to compare medians. The Mann–Whitney test was used to compare two groups and Kruskal–Wallis test was used to compare three or more groups. A *post hoc* Dunn test was used to compare multiple independent variables. Spearman analysis was applied to correlate TSH and UIC levels. Proportions were analyzed with 2 × 2 contingency tables and Fisher’s exact test. A logistic regression analysis was performed using TSH concentration as dependent variable and UIC, weight status, WC, sex, and age as independent variables.

Microsoft Excel 2007 (Microsoft Corporation, Redmond, WA, USA), SPSS 16.0 (SPSS Inc., Chicago, IL, USA), and GraphPad Prism 9.0.0 (Graph Pad Software Inc., La Jolla, CA, USA) were used to perform statistical analyses.

## 3. Results

We evaluated a total of 307 schoolchildren, of whom 274 were included for the final analysis. Thirty-three children were excluded due to incomplete data or unusable samples.

Table 1 shown the general characteristics of schoolchildren. Considering all participants, 49.6% were female, with a median age of 9 years and an IQR of 7–10 years. Median of UIC was 442 μg/L with and IQR of 290–671 μg/L. Comparison of medians between girls and boys did not show statistically significant differences. On the other hand, the median of TSH was 4.1 μUI/mL with an IQR of 3.8–4.5 μUI/mL. Comparison of medians between female and male showed a significant difference, with a higher TSH level in girls than in boys [4.1 (IQR 3.8–4.7) vs. 4.0 (IQR 3.8–4.3); *p* = 0.0133]. 

We found a global prevalence of 40.5% of overweight and obesity, without a significant difference between girls and boys. Nonetheless, the evaluation of WC < 10 pctl, was different, with higher prevalence in males. We also compared TSH levels with weight status (underweight, normal weight, overweight, obese) with no significant differences.

Table 2 shows the comparison of TSH levels and UIC between all ages of schoolchildren. UIC in 7-year-old children was higher compared with those 11 years old (*p* < 0.05). There were no significant differences in UIC in TSH levels at any age.

UIC and TSH levels according to weight status and WC showed no significant differences among different categories (Table 3).

Fisher’s exact test was used to analyze associations between UIC and TSH levels in schoolchildren. Values of UIC above median (442 μg/L) increase significant the risk of TSH ≥ 5 μUI/mL (OR 2.4, CI95% 1.0–5.52) when this was made with other UIC levels (</≥ 100 μg/L and </≥ 300 μg) we did not find this association significant (Table 4).

The logistic regression found significant association between UIC and TSH levels (*p* < 0.05) with three different models: model 1 (adjusted for age and sex), model 2 (adjusted for age, sex, and WC), and model 3 (adjusted for age, sex, and weight status) (Table 5).

## 4. Discussion

This study was the first to our knowledge that evaluated UIC, TSH levels and nutritional status in schoolchildren from northeastern Mexico. We found high prevalence of overweight and obesity, and an excessive high iodine intake in schoolchildren. This data agrees with the Mexican Health and Nutritional Survey that reports a 35.5% national prevalence between overweight and obesity in schoolchildren aged 5 to 11 years, and a total of 42.9% and 34.8% between overweight and obese boys and girls from northern cities of Mexico, respectively (ENSANUT 2018–2019) [10]. Moreover, iodine intake also agrees with previous studies in schoolchildren from the urban population of Central Mexico [9].

In this study we assessed TSH levels from dry blood samples from schoolchildren. This indicator was useful to determine global TSH levels, but it also could be useful for individual assessments; we found 26 children (9.5%) with TSH levels ≥ 5 μUI/mL, this finding was similar to a previous report of Sánchez RLM, et al. 2012 [17], who reported 10.9% prevalence of subclinical hypothyroidism in children aged 2–12 years in northern Mexico, considering a TSH ≥ 4.5 μUI/mL to ≤10 μUI/mL as the diagnostic range; this prevalence was the highest found compared to the other regions of Mexico [17], despite trivial differences in age ranges and cut-off values for TSH. We found two schoolchildren with TSH > 8 μUI/mL, a 10-year-old boy with TSH of 8.3 μUI/mL and a 7-year-old girl with TSH of 10.2 μUI/mL. No TSH results were suggestive of hyperthyroidism. The median of TSH of 4.1 μUI/mL differs from other studies in Mexican population. Mendez et al.; 2016 reported a median of serum TSH of 2.06 (IQR 1.12–2.76 μUI/mL) and 2.0 (IQR 1.52–3.26 μUI/mL) in children aged between 6–12 years old, with normal weight and obesity, respectively [9]. In this study we also analyzed the median of UIC and TSH levels with different nutritional status and we did not find significant differences between groups. 

We did not find correlations between WC and TSH levels. Interestingly, Shalitin et al.; 2009 evaluated prevalence of thyroid dysfunction in obese children before and after weight reduction and its relation with metabolic parameters such as WC; they found an association of hyperthyrotropinemia (TSH ≥ 5 μUI/mL) with higher WC in children and adolescents aged 5–18 years, with a positive correlation between the decrease in TSH levels and the decrease in WC, considering hyperthyrotropinemia with TSH levels ≥ 4.0 μUI/mL [18].

We found a significant association between UIC above of median (442 µg/L) and TSH levels ≥ 5 µUI/mL, although this association was not present when UIC was ≥300 µg/L. As mentioned in 1948 by Wolff and Chaikoff, organic binding of iodine within the gland, can be almost completely blocked by raising the level of plasma inorganic iodine above a certain critical level; as soon as the level of plasma inorganic iodine falls below its critical range, the gland again resumes the iodine uptake. This regulator probably serves to prevent the formation of excessive amounts of hormone by the gland when the body is suddenly flooded with iodine [19]. The mechanism for the acute Wolff–Chaikoff effect is not completely understood, but is thought to be at least partially explained by the generation of several inhibitory substances (such as thyroidal iodolactones, iodoaldehydes and/or iodolipids) due to the thyroid peroxidase activity [5]. In addition, the Wolff–Chaikoff effect may also present an escape effect, due to a decrease in the expression of the sodium iodide symporter (NIS) [20]. As established, the decrease in TH synthesis due to the Wolff–Chaikoff effect can be transient (24–48 h) [21]. Previous reports of Japanese population demonstrated the same TSH behavior with high UIC, as well as a TSH decrease to normal levels in some patients with only iodine dietary restriction [22,23,24]. We measured TSH once, without any dietary restriction.

Our study had some limitations, the number of children studied was not equally distributed among the three schools. On the other hand, a complete thyroid function test was not performed in our participants, we only measured TSH on Guthrie cards, although there are no specific standard values for TSH evaluation in these cards for school-aged children. It would also have been interesting to correlate plasma glucose and insulin levels with somatometry of all participants. Finally, we did not measure Tg, though, it is a marker used in iodine evaluation. High Tg levels have been considered a sensitive indicator of low and high UIC > 300 μg/L concentrations in the schoolchildren population aged between 6 to 12 years old from different countries [25]. Further studies are needed in our population to evaluate Tg behavior and its association with UIC variations.

## 5. Conclusions

In conclusion, this study found higher UIC levels (442 μg/L) than those previously reported in a national estimation in Mexico (297 μg/L; 2011) [3]. We found an association between high UIC and TSH levels, but we did not find associations between UIC levels and WC. Additional studies are needed to evaluate the causes and health outcomes in this population.

## Figures and Tables

**Table 1 nutrients-13-03975-t001:** General characteristics of schoolchildren of northeastern Mexico.

	All (*n* = 274)	Female (*n* = 136)	Male (*n* = 138)	*p* Value
Age (yr) *	9 (7–10)	9 (7–10)	9 (7–10)	0.3669
UIC ( μg/L ) *	442 (290–671)	444 (265–669)	432 (293–682)	0.3595
TSH ( μUI/mL ) *	4.1 (3.8–4.5)	4.1 (3.8–4.7)	4.0 (3.8–4.3)	0.0133
Prevalence (%) of TSH > 5 ( μ UI/mL) **	5.3 (5.0–6.1)	5.1 (5.0–6.0)	5.9 (5.1–7.8)	0.1038
TSH ( μ UI/mL), 90th pctl	4.9	5.0	4.8	-
TSH ( μ UI/mL), 95th pctl	5.3	5.4	5.2	-
Weight status **				
Underweight, *n* (%)	12 (4.4%)	5 (3.7%)	7 (5.1%)	0.3948
Normal weight, *n* (%)	151 (55.1%)	78 (57.4%)	73 (52.9%)	0.2678
Overweight, *n* (%)	51 (18.6%)	29 (21.3%)	22 (15.9%)	0.1613
Obese, *n* (%)	60 (21.9%)	24 (17.6%)	36 (26.1%)	0.0612
Waist circumference **				
<10th pctl, *n* (%)	25 (9.1%)	5 (3.7%)	20 (14.5%)	0.0015
10th–90th pctl, *n* (%)	188 (68.6%)	100 (73.5%)	88 (63.8%)	0.0535
>90th pctl, *n* (%)	61 (22.3%)	31 (22.8)	30 (21.7%)	0.4742

* Median and interquartile ranges (IQR) 25th–75th pctl and median compared by sex using the Mann–Whitney U Test. ** Comparison of proportions by sex were analyzed using the Fisher’s exact test. A *p* < 0.05 was considered significant.

**Table 2 nutrients-13-03975-t002:** Comparison of TSH levels and UIC between ages in schoolchildren of Northeastern Mexico.

Age (yr)	UIC (μg/L) *	TSH (μUI/mL) *
5–6 (*n* = 31)	399 (253–700) ^a,b^	4.2 (3.9–4.8) ^a^
7 (*n* = 56)	341 (140–522) ^b^	4.1 (3.8–4.6) ^a^
8 (*n* = 36)	494 (259–694) ^a,b^	4.1 (3.7–4.4) ^a^
9 (*n* = 67)	415 (292–596) ^a,b^	4.0 (3.8–4.4) ^a^
10 (*n* = 57)	293 (369–707) ^a^	4.0 (3.8–4.7) ^a^
11 (*n* = 27)	613 (436–805) ^a^	4.2 (3.8–4.5) ^a^

* Median and interquartile range (IQR) 25th–75th. A Kruskal–Wallis test with a *post–hoc* Dunn test was performed, “a” and “b”superscripts (corresponding to 7, 10 and 11 years old) show significant differences (*p* < 0.05), meanwhile “a,b” superscripts (corresponding to 5–6, 8 and 9 years old) show not statistically differences.

**Table 3 nutrients-13-03975-t003:** Comparison UIC and TSH levels according with different categories of weight status and waist circumference.

	UIC (μg/L)	TSH (μUI/mL)
Weight Status		
Underweight (*n* = 12)	528 (409–676)	4.2 (3.8–4.6)
Normal weight (*n* = 151)	412 (257–659)	4.1 (3.8–4.5)
Overweight (*n* = 51)	408 (246–649)	4.1 (3.9–4.4)
Obese (*n* = 60)	529 (348–773)	4.0 (3.7–4.7)
*p* value	>0.05	>0.05
Waist Circumference		
<10th pctl	484 (251–603)	4.2 (3.8–4.6)
10th–90th pctl	437 (288–675)	4.1 (3.8–4.5)
>90th pctl	455 (293–692)	4.0 (3.8–4.7)
*p* value	>0.05	>0.05

Weight Status: CDC BMI charts; Waist circumference: IDF consensus of Metabolic Syndrome in Children and Adolescents—Mexican American population. Comparisons were performed with the Kruskal–Wallis test and the *post–hoc* Dunn test.

**Table 4 nutrients-13-03975-t004:** Association between UIC and TSH levels in schoolchildren of northeastern Mexico.

	TSH ≥5 μUI/mL	TSH <5 μUI/mL	OR	CI 95%	*p* Value
UIC < 100 (μg/L)	2 (0.7%)	13 (4.7%)	1.5	0.3–6.5	0.4267
UIC ≥ 100 (μg/L)	24 (8.8%)	235 (85.8%)			
UIC ≥ 300 (μg/L)	21 (7.7%)	179 (65.3%)	1.6	0.6–4.0	0.2450
UIC < 300 (μg/L)	5 (1.8%)	69 (25.2%)			
UIC > median (μg/L)	18 (6.6%)	119 (43.4%)	2.4	1.0–5.5	0.0309
UIC < median (μg/L)	8 (2.9%)	129 (47.1%)			

UIC median value was 442 μg/L. A Fisher exact test was performed to analyze 2 × 2 contingency tables. A *p* < 0.05 was considered significant.

**Table 5 nutrients-13-03975-t005:** A UIC value above 442 μg/L increases levels of TSH ≥ 5 μUI/mL in schoolchildren of northeastern Mexico.

	OR	CI 95%	*p* Value
Model 1	2.583	1.058–6.308	0.037
Model 2	2.813	1.134–6.976	0.026
Model 3	2.436	0.990–5.995	0.053

A logistic regression was applied according with the following models: Model 1 adjusted for age and sex, Model 2 adjusted for age, sex, and WC, and Model 3 adjusted for age, sex, and weight status. A *p* < 0.05 was considered significant.

## Data Availability

Data available on request due to restrictions (e.g., privacy or ethical principles).

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
