# Peer review of "High Iodine Urinary Concentration Is Associated with High TSH Levels but Not with Nutrition Status in Schoolchildren of Northeastern Mexico"

_nutrients, 2021, doi:10.3390/nu13113975_

Round 1
Reviewer 1 Report
This is an interesting work in which the authors evaluate the UIC and TSH levels in a sample of ca. 300 schoolchildren from Northeaster Mexico. Moreover, they also study the possible correlation between UIC / TSH levels and different factors, such as BMI and waist circumference, finding no significant relation. However, they find an interesting correlation between UIC and TSH.
I think that the article can be accepted almost in its current form. However, before the final acceptance of the article, I would suggest the application of very few changes:
- Though the abbreviation TSH is probably well-known by the community to which the article is addressed, it would be worth to mention (at least in the abstract) the complete name “thyroid-stimulating hormone”.
- I think that evaluation of Tg levels would be a measure that would result of great interest for the study, as well as the determination of its correlation with UIC and TSH levels. I would be possible to measure it? Are the samples still available?
Author Response
Point 1. Though the abbreviation TSH is probably well-known by the community to which the article is addressed, it would be worth to mention (at least in the abstract) the complete name “thyroid-stimulating hormone”.
Response: Thank you for your encouraging comments, following reviewer's recommendation, we have added in the abstract and in the introduction, the full name of the thyroid stimulating hormone before its TSH acronym (page 1, line 22 and page 2 line 55, respectively).
Point 2. I think that evaluation of Tg levels would be a measure that would result of great interest for the study, as well as the determination of its correlation with UIC and TSH levels. I would be possible to measure it? Are the samples still available?
Response: We agree with this interesting comment, thyroglobulin measurement would complement our work. Unfortunately, we have not yet implemented the technique to measure thyroglobulin on Guthrie cards. In addition, as far to our knowledge, this study is not available in the country. Measurement of thyroglobulin requires further investigations, which we definitely will consider in future works, thank you so much for you valuable suggestion.
Reviewer 2 Report
This is an interesting article, however, I think you may introduce some modifications.
- In abstract: what is CC? (line 29) Please, explain the abbreviation.
- I think, in the introduction, you can discuss the substances, in food, which impaired iodine absorption.
- You recruited children from northeastern Mexico, but previously you discussed a study that referred to iodine status in Mexico. Can you refer to data about iodine status in the region, where you did research?
- Why did you not assess a T4 or/and T3 levels? Do you think, that TSH is better for evaluate thyroid function, especially among overweight and obese children?
- Do you ask about supplementation? Including iodine supplement, of course.
- You could introduce more information about nutritional sources of iodine.
Good luck
Author Response
We are pleased to submit the reviewed version of the original article entitled “ High Iodine Urinary Concentration is Associated with High TSH Levels but no with Nutrition Status in Schoolchildren of Northeastern Mexico” by the authors for consideration for publication in ¨Nutrients¨.
In the new version of the manuscript, we did our best effort to incorporate all the reviewer’s comments and suggestions. The point-by-point responses to reviewers are attached below.
Thank you for your consideration.
Sincerely,
Dr. Augusto Rojas-Martinez, Dr. Aidy González-Núñez
Response to Reviewer 2 Comments
Point 1. This is an interesting article, however, I think you may introduce some modifications. In abstract: what is CC? (line 29) Please, explain the abbreviation.
Response: we appreciate reviewer´s observation, we misspelled ¨C¨ for ¨W¨; this has already been changed, ¨WC instead of CC¨ (page 1, line 29). To clarify, the term waist circumference was added (lines 25-26), where the term was initially referred, followed by the correct acronym (WC).
Point 2. I think, in the introduction, you can discuss the substances, in food, which impaired iodine absorption.
Response: Thank you for your suggestion. In the new version of the manuscript, we added relevant references with information about food and substances (some of them contained in food), that may impair iodine absorption and can interact with thyroid function (page 2, lines 60-64).
Point 3. You recruited children from northeastern Mexico, but previously you discussed a study that referred to iodine status in Mexico. Can you refer to data about iodine status in the region, where you did research?
Response: This is a very interesting question. To our knowledge, there is no data of iodine status in the region where we did the study; therefore, national estimations in the country may have some subestimations. This would be the first time this population is evaluated.
Point 4. Why did you not assess a T4 or/and T3 levels? Do you think, that TSH is better for evaluate thyroid function, especially among overweight and obese children?
Response: Thank you for your enquiry. Undoubtedly, T4 and/or T3 would have complemented our investigation, especially, considering that we found a high prevalence of overweight and obese children. Unfortunately the technique to meaure thyroid hormones on Guthrie cards, aside of TSH is not available. Nevertheless, we evaluated the association of TSH and BMI changes, with no statistically results. We appreciate your enriching suggestion, we definitely will take it into account for further investigations.
Point 5. Do you ask about supplementation? Including iodine supplement, of course.
Response: Thank you for your interest. As part of our exclusion criteria, we thoroughly asked for iodine supplementation, as well as for prescriptions or any preexisting condition that could alter UIC and/or TSH results. We included this specification in Material and Methods (page 3, line 93).
Point 6. You could introduce more information about nutritional sources of iodine.
Response: We appreciate your valuable comments. In the new version of the manuscript, we added more information about nutritional sources of iodine, with its references (page 2, line 59). This complements previous data about Mexican intake of iodine rich food (page 2, lines 69 - 71).

Round 2
Reviewer 2 Report
No comments.